# Changing the Recipe: Pathogen Directed Changes in Tick Saliva Components

**DOI:** 10.3390/ijerph18041806

**Published:** 2021-02-12

**Authors:** Michael Pham, Jacob Underwood, Adela S. Oliva Chávez

**Affiliations:** Department of Entomology, College Station, Texas A&M University, TX 77843, USA; mpham@tamu.edu (M.P.); jacob.underwood@tamu.edu (J.U.)

**Keywords:** saliva-assisted transmission, *Anaplasma phagocytophilum*, tick-borne encephalitis, *Borrelia burgdorferi*, ticks, tick saliva, tick-borne diseases

## Abstract

Ticks are obligate hematophagous parasites and are important vectors of a wide variety of pathogens. These pathogens include spirochetes in the genus *Borrelia* that cause Lyme disease, rickettsial pathogens, and tick-borne encephalitis virus, among others. Due to their prolonged feeding period of up to two weeks, hard ticks must counteract vertebrate host defense reactions in order to survive and reproduce. To overcome host defense mechanisms, ticks have evolved a large number of pharmacologically active molecules that are secreted in their saliva, which inhibits or modulates host immune defenses and wound healing responses upon injection into the bite site. These bioactive molecules in tick saliva can create a privileged environment in the host’s skin that tick-borne pathogens take advantage of. In fact, evidence is accumulating that tick-transmitted pathogens manipulate tick saliva composition to enhance their own survival, transmission, and evasion of host defenses. We review what is known about specific and functionally characterized tick saliva molecules in the context of tick infection with the genus *Borrelia*, the intracellular pathogen *Anaplasma phagocytophilum*, and tick-borne encephalitis virus. Additionally, we review studies analyzing sialome-level responses to pathogen challenge.

## 1. Introduction

Ticks are capable of transmitting a wide variety of pathogens including viruses, bacteria, protozoans, fungi, and nematodes of medical and veterinary importance [1]. They are obligatory blood-feeding arthropods that are divided into three families: Argasidae (soft ticks), Ixodidae (hard ticks), and Nuttallielidae [2,3]. Soft ticks feed repeatedly for a short period (minutes to hours), while hard ticks feed for several days to weeks, depending on the life stage [4]. Ixodid larvae and nymphs take up to eight days to complete the bloodmeal, whereas adult females can feed for up to 12 days or longer [5]. Furthermore, unlike mosquitoes that pierce through the skin to reach blood vessels, ticks produce a feeding pool by cutting through the host skin with their chelicerae. The chelicerae extend and lacerate the epidermis, which is then followed by the insertion of the hypostome into the dermis [6], producing significant damage. The relatively long period of feeding and the type of lesion require the inhibition of host immunity and localized hemostasis. 

Vertebrate skin represents a key environmental interface acting as a physical and immune barrier that is composed of two layers: Epidermis and dermis. These layers contain immune cells and effectors that together comprise a complex network of inflammatory, innate, and acquired immune defense mechanisms [7,8]. Keratinocytes act as sentinels detecting a pathogen associated molecules and toxins [9,10]. These cells interact with T cells to maintain tissue homeostasis and coordinate immune responses [11]. The wound healing response (including hemostatic plug formation, vasoconstriction, inflammation, and tissue remodeling) as well as pain and itch sensations occur in response to injury. Platelets also act as immune sentinels of damaged vessels. They guide neutrophils and other leukocytes to the site of extravasation and direct innate and adaptive immune responses [12,13]. Other immune cells residing in the skin such as Langerhans and dermal dendritic cells (DC) migrate to the lymph nodes and activate inflammatory and immune responses upon encountering antigens [14,15]. To counteract these challenges, ticks have evolved and acquired several effectors that diminish these immune and hemostatic responses.

Specifically, tick saliva delays wound healing and blood clotting as well as inflammatory responses with secreted molecules that interact with cytokines, chemokines, and growth factors [16,17]. Additionally, tick salivary glands release a wide number of immunomodulatory and anti-hemostatic molecules. These components maintain blood flow and reduce itching, inflammation, and immune rejection at the skin interface with the attached tick, allowing blood feeding to succeed. For example, tick saliva contains inhibitors (thought to include Angiotensin-converting enzyme (ACE) metalloproteases, endocannabinoids, adenosine, and others) that reduce pain and itching, preventing physical tick removal by the host [18,19,20]. Tick saliva also inhibits the migration of neutrophils [21] and macrophages [22] and can alter dendritic cell activation and function [23,24]. Through these bioactive components, tick saliva creates an immune-privileged local environment at the bite site that facilitates pathogen transmission. Therefore, it is not surprising that tick-borne pathogens take advantage of tick salivary secretions to enhance their establishment and infection. Herein, we review how tick-borne pathogens exploit and augment the immunomodulatory and regulatory properties of tick saliva, focusing specifically on *B*. *burgdorferi*, *Anaplasma phagocytophilum*, and tick-borne encephalitis virus (TBEV). 

## 2. Effect of Saliva on Pathogen Transmission 

### Saliva Assisted Transmission (SAT)

Saliva-assisted transmission (SAT) has been observed with several tick-borne pathogens [3,25]. Several studies have reported enhanced vertebrate infection by tick-borne pathogens after needle injection along with salivary gland extracts (SGE) as compared to pathogens alone. Nevertheless, the mechanisms and molecules involved in this process have not been completely characterized. Only a relatively small number of salivary components associated with tick-borne pathogen transmission have been described, some of which are listed in Table 1 and Table 2. Herein, we will discuss how sialostatin L and L2 enhance the transmission of *A*. *phagocytophilum*, TBEV, and *B*. *burgdorferi*. Additionally, salivation has also been associated with pathogen transmission in non-vertebrate vector-host systems such as *Varroa destructor* and *Apis mellifera*, where salivation alters hemocyte behavior as well as in plant diseases caused by whiteflies, aphids, mites, and psyllids, where vector salivation plays a role in altering host plant defenses [26,27,28,29,30,31,32,33].

Tick saliva is likely to influence pathogen acquisition. Reppert et al. [34] showed that tick feeding results in an increased number of neutrophils at the bite site of *A*. *phagocytophilum* infected and uninfected sheep. Interestingly, infection also appears to augment the number of neutrophils even in the absence of feeding ticks. Immunohistochemistry (IHC) experiments demonstrated the presence of infected neutrophils at the bite site. Infected neutrophils at the bite site have been previously reported in naturally infected lambs [35]. It is highly possible that components in tick saliva and the inflammation at the bite site results in the chemoattraction of these infected neutrophils. Chemoattraction of tick SGE has been shown for *B*. *burgdorferi.* This mechanism may explain the phenomenon of *B*. *burgdorferi* acquisition between co-feeding infected and uninfected ticks [36]. However, the exact molecules that facilitate this chemoattraction are not known. 

Sialostatins are C1-type cysteine protease inhibitors (cystatins) that suppress the action of mammalian cathepsins [37,38]. Cathepsins L and S play important roles in major histocompatibility complex (MHC) II antigen processing and presentation by cells in the cortical epithelium of the thymus and professional antigen presenting cells (APCs), respectively [39]. These proteins also play a role in the suppression of chemokines, such as IP-10 (CXCL10), MIP-2 (CXCL2), MCP-1 (CCL2), RANTES (CCL5), LIX (CXCL5), CXCL16, MIP-1β (CCL4), and MIP-1α (CCL3), and cytokines including TNFa, IL-9, IL-1β, and IL-12 [40,41,42,43]. Sialostatin L decreases the activation of interferon regulatory factor 4 (IRF4) signaling in mast cells [40] and JAK/STAT signaling in DCs by diminishing phosphorylation of STAT-1 and STAT-2 [42]. Additionally, Sialostatin L dampens antigen mediated CD4+ proliferation [43]. Thus, these proteins affect both innate and adaptative immune responses, which in turn impacts pathogen colonization in the host. For example, sialostatin L2 reduces inflammasome activation by targeting caspase 1, affecting cytokine secretion and inflammatory responses during *A*. *phagocytophilum* infection [44]. Tick-borne encephalitis virus (TBEV) replication in DCs is enhanced in the presence of sialostatin L2, by diminishing the antiviral effect of IL-1β [42]. Likewise, these cystatins decrease DC activation after *B*. *burgdorferi* infection by interfering with Erk1/2 signaling [41]. These examples demonstrate the particular impact of these two salivary proteins on pathogen establishment. Other proteins have also been shown to positively impact pathogen transmission, with evidence to show that vector saliva can increase pathogen recruitment to the feeding site, explaining the evolutionary advantage that SAT represents for tick-borne pathogens [34].

## 3. Role of Tick Salivary Components during Tick Feeding and Pathogen Transmission

### 3.1. Effects on Hemostasis and Angiogenesis 

Hemostasis is described as the balance of physiological processes that maintain blood flow and fluidity while preventing excessive blood loss at the site of a vascular injury [45] and is thus part of the wound healing responses. As the first step in wound healing, hemostasis includes vasoconstriction, followed by two linked processes: Primary hemostasis, which involves platelet aggregation, and secondary hemostasis, which induces the coagulation cascade. The activation of either the extrinsic or intrinsic coagulation pathways leads to the activation of Factor X. Activated Factor X (Factor Xa) eventually drives the conversion of prothrombin to thrombin. Crosslinked fibrin binds to the aggregated platelet plug, forming the thrombus, which stops bleeding. Wound restoration starts by the formation of new connective and granular tissue by a process of re-epithelization and neovascularization during angiogenesis [46]. 

Tick saliva promotes continuous blood flow with molecules that counteract the different hemostatic steps and processes involved in angiogenesis (Table 1). Some of these effectors include Salp14 and Iris identified in *I*. *scapularis* and *I*. *ricinus,* respectively [47,48] (Figure 1). Salp14, a 28 kDa protein, delays blood coagulation by specifically inhibiting Factor Xa [47]. Iris, on the other hand, is a serpin (serine protease inhibitor) with anti-coagulant, anti-hemostatic, and immunomodulatory properties [48]. It inhibits human leukocyte elastase (by ~70%), thrombin tissue plasminogen activator (tPA), and Factor Xa (by around 30%). This serpin significantly delays the intrinsic coagulation pathway and fibrinolysis, affecting blood clot formation [48] and dampening wound healing responses (Figure 1). 

The effects of these molecules on pathogen transmission have not been investigated, with the exception of the tick histamine release factor (tHRF) and *B*. *burgdorferi* (Table 1) [49]. Although there are few known examples of tick-borne pathogens directly exploiting tick proteins involved in delaying hemostasis and angiogenesis, it is likely that constant blood flow and delay in wound closure benefit tick-borne pathogen transmission. A list of proteins, their targets, effect, and whether they facilitate pathogen transmission experimentally is described in Table 1 [47,49,50,51,52,53,54,55,56,57,58,59,60,61,62,63,64,65,66,67,68,69,70,71,72,73,74,75,76,77,78,79,80].

### 3.2. Immunomodulatory Components: Effects on Host Defenses

#### 3.2.1. Host Defense 

Part of the wound healing response is the activation of inflammation, which recruits immune cells into the site of tissue damage. Damage associated molecular patterns (DAMPs), increases in intracellular Ca++, chemokines, and lipid mediators lead to the recruitment of immune cells, such as neutrophils, macrophages, mast cells, T cells, and other cells. These cells prevent infection and assist generating growth factors that lead to tissue repair [46]. Ticks have evolved several strategies to evade these host immune responses in order to ensure feeding to repletion and further development [7,16,82,83,84]. The capability of ticks to attach for a prolonged time has stimulated strong interest in investigating tick feeding. A necessary step in understanding tick feeding includes defining the molecular composition of tick saliva, which is also important for understanding the role of saliva in pathogen transmission. The use of global transcript and protein profiling, as well as comparative analyses, has led to the discovery of several molecules required for the induction and modulation of immune defenses [25]. However, while the general effects of tick-directed immunomodulation are known, the characterization of tick salivary gland molecules with regulatory functions is still relatively limited. Some of these molecules and the tick species that secrete them are identified in Table 2 [16,21,22,23,24,38,42,49,68,72,85,86,87,88,89,90,91,92,93,94,95,96,97,98,99,100,101,102,103,104,105]. 

#### 3.2.2. Complement

The complement system is a series of protein cascades that function as part of the innate immune response, recognizing damaged tissue and microbial invaders. Three complement pathways have been characterized: The classical, lectin, and alternative pathway. Complement is always active at low levels, with regulators active to control the response [106]. When pattern recognition molecules, such as C1q, antibodies, and pentraxins, bind to foreign molecules, microbes, apoptotic, or damaged cells, other proteins in the complex are activated leading to the amplification of the cascade. This cascade results in lysis of the cell through the creation of pores in the cell’s membrane by the membrane attack complex. Other outcomes from the activation of the complement pathways include opsonization, phagocytosis, B cell activation, T cell regulation, and the activation of inflammatory signaling [106,107,108,109]. The complement pathways act as a sentinel and control during tissue damage and pathogen invasion. Thus, it is not surprising that ticks have evolved several effector proteins that target components in the complement cascade, specifically C3 and C5 (Table 2), which benefit pathogen transmission. 

Members of the *B*. *burgdorferi*
*s*.*s*. and *s*.*l*. complexes are the causative agents of Lyme disease in the US and Europe. They are gram-negative spirochete bacteria, transmitted by Ixodid ticks. *Borrelia* spp. pathogens are some of the tick-borne pathogens that specially benefit from the dampening of the complement pathway. *Borrelia* spp. pathogens activate all complement pathways, even in the absence of *Borrelia* specific antibodies [110]. Therefore, they have acquired several mechanisms by which they can escape complement killing, including the exploitation of tick salivary components. For example, Salp20 and Isac both inhibit complement activity, enhancing *B*. *burgdorferi* transmission [96,104,105]. TSLPI (tick salivary lectin pathway inhibitor), interferes with complement activation by binding to the active sites of mannose binding lectin, a C-type lectin that detects oligosaccharides. This protein also hinders the phagocytosis of the bacteria by neutrophils and the rest of the cascade that finalizes with the membrane attack complex [111]. Furthermore, *B*. *burgdorferi* increases transcription of this protein in the salivary glands of nymphs [111], possibly to increase its survival. A homolog to this protein was identified in *I*. *ricinus*, and it protects both *B*. *burgdorferi*
*s*.*s*. and *B*. *garinii* from complement killing during in vitro tests [112]. Other tick-borne pathogens may also take advantage of these effector to survive the mammalian complement cascade.

#### 3.2.3. Immune Cells

Mechanical abrasion of the skin leads to the activation of skin immune cells that prevent the entry and establishment of pathogenic invaders into the body [113]. The immune response is mediated by specialized and non-specialized cells that have evolved to recognize non-self-antigens. Non-specialized cells, such as endothelial cells, keratinocytes, fibroblasts, and platelets communicate with specialized innate immune cells, like macrophages, DCs, neutrophils, Langerhans cells, mast cells, basophils, γδ T cells, and innate lymphoid cells (ILCs) [113,114]. These cells are largely responsible for the great majority of the immune response, from phagocytizing microbes and damaged cells as well as producing the effectors necessary for a coordinated immune response. For example, DCs with bound antigens migrate from the skin to the lymph nodes to present them to B and T cells, which in turn undergo maturation and clonal replication. In mice, and possibly humans, DCs and macrophages can interact with T cells in the skin during inflammation, providing a site where antigen presentation can occur. This site is termed inducible skin-associated lymphoid tissue (iSALT) and is key in the generation of adaptive immunity [114]. Phagocytes at the bite site will begin to engulf damaged tissue and invaders they may encounter in the area. Granular cells release a cocktail of compounds into the extracellular space, prepping the somatic cells for the inflammatory response, and activating other immune cells [7,115,116]. This complex of immune reactions and cells profoundly influences tick feeding efficiency and the outcome of pathogen transmission.

The tick encounters these defense systems during feeding, and some animal species may acquire resistance to tick feeding after infestation by generating systemic immunity [117]. This immunity has been reported in animals such as cattle, guinea pigs, and rabbits. [117,118,119]. This immunity has also been shown to have some effect between tick species on a single host [120]. Several immune cells, particularly basophils and resident memory T cells, have been linked with the development of this resistance. Acquired tick immunity (AIT) can disrupt the ability of ticks to complete a blood meal, reproduce, and even transmit pathogens. The effects of cellular immunity on tick physiology has resulted in evolutionary pressures for ticks to develop several molecules that can counter immune cell activation, migration, and proliferation (Table 2). One example of such immunomodulatory proteins is the serpin Iris (Figure 1). Iris is an immunosuppressant that affects T cell proliferation and cytokine secretion by macrophages, promoting a Th2 response with the generation of high antibody titers that by themselves are ineffective in controlling infections with tick-borne pathogens [97,121,122].

Tick-borne pathogens take advantage of these molecules by increasing their expression and may use them as a protective coat. *B*. *burgdorferi* selectively enhances Salp15 expression in SGs and directly binds Salp15 through OspC, a spirochete surface protein [99]. Salp15 and its orthologs have also been shown to inhibit the activation of CD4+ T-lymphocytes and keratinocytes [123,124,125] (Figure 1). BIP (B-cell inhibitory protein), identified from *I*. *ricinus* SGs, inhibits OspC-induced B lymphocyte proliferation [91]. Therefore, facilitating *B. burgdorferi* infection. Similarly, *A*. *phagocytophilum*, a gram-negative obligate intracellular tick-borne bacterium that colonizes polymorphonuclear neutrophils, increases the expression of Salp16, where it is required for the initial infection of the salivary gland [125]. This protein is involved in the inhibition of neutrophils and decreased reactive oxygen species (ROS) production [21]. However, whether *A*. *phagocytophilum* uses this protein for its transmission has not been defined.

#### 3.2.4. Cytokine and Chemokine Secretion 

Cytokines are a collection of effector molecules, modulating innate and adaptive immune responses through a network of complex and at times, contradictory interactions. These proteins interact with immune cells and neurons to coordinate immune responses, including the inflammation of tissues, aggregation of immune cells and somatic cells, proliferation of immune cells, cell recruitment, T cell differentiation, maturation of B-cells, and itch [126,127,128]. Itch sensations if left unchecked, would lead to injury awareness and to the host scratching or grooming, leading to tick dislodgement. Ticks dampen itch sensations through secreted salivary components such as lipocalins that bind histamine and degrade bradykinin, mediators of pain and itch [20,25]. Chemokines are 8–12 kD molecules that induce a chemotaxis of various immune cell types, including neutrophils, monocytes, lymphocytes, eosinophils, T and B cells fibroblasts, and keratinocytes [126,127]. Keratinocytes and other cells in the skin secrete cytokines upon infection or damage [129]. Platelets also express several chemokine receptors that activate their migration, aggregation, and granule release [130]. Thus, cytokines and chemokines play a crucial role in the response against ticks and tick-borne pathogens. However, ticks secrete several effectors that affect the expression and regulation of cytokines and chemokines at the bite site (Table 2). IRS-2, for example, diminishes the secretion of IL-6 and IL-17 and the development of Th17 helper T cells [23], which are an important subset of T cells found in the skin. IL-17 has been associated with the production of antibodies against *B*. *burgdorferi* during early Lyme disease [128]. Therefore, the dampened Th17 T cell development and IL-17 secretion likely benefits *B*. *burgdorferi* early infection and establishment in the skin. 

#### 3.2.5. Histamine Secretion

Immune cells secrete other immune effectors, including histamine. Histamine influences the polarization of immune responses and the maturation of immune cells [131]. Furthermore, histamine is an important effector secreted by basophils and mast cells in response to tick bites and has been associated with anti-tick immunity [117]. However, although ticks secrete several histamine binding proteins that may block the effect of some of the histamine at the bite site, it appears certain levels of it benefit tick feeding and pathogen transmission. tHRF is a protein that has been found in *I*. *scapularis* and *D*. *andersoni* saliva [49,69]. This protein binds to basophils, inducing histamine release and promoting vasodilation. Interestingly, this protein is upregulated by *B*. *burgdorferi* infection and is required for the efficient transmission of this pathogen. While seemingly detrimental to the tick, and potentially the pathogen, by increasing the blood flow into the bite site, pathogens may facilitate their own dissemination in the host [49].

## 4. Global Manipulation of Tick Sialome by Pathogens

The sialome of the tick consists of mRNAs and proteins expressed in the salivary glands [132]. Global studies on tick salivary gland gene expression have identified over 287,000 transcripts from which over 45,000 putative secretory proteins have been identified [133]. However, only a small proportion (~5%) of salivary gland proteins have had their predicted functions verified [8]. In order to be transmitted to the vertebrate host, tick-borne pathogens must first invade the tick salivary glands where they manipulate the sialome. Several studies have demonstrated the ability of tick-borne pathogens to change the gene expression in tick salivary glands. We will focus on three pathogens: *A*. *phagocytophilum*, *B*. *burgdorferi* complex, and TBEV, and the processes that they manipulate to facilitate their transmission.

A proteomic screening revealed that *A. phagocytophilum* increases the expression of anticlotting proteins, immune inhibitor proteins, and prolyl 4-hydroxylase subunits in *I*. *scapularis* salivary glands [134] (Figure 2A). The anticlotting factors include Salp9 [47] and Salp11 [135] and thrombin inhibitors. Another anticlotting factor, Metis-1, is upregulated during *Anaplasma* infection. Metis-1 is a salivary gland specific to metalloprotease thought to stimulate fibrinolysis [56]. Immune inhibitors, Sialostatin L and Sialostatin L2, also have increased protein levels [134]. Multiple subunits of the proline 4-hydroylase enzyme are upregulated in salivary glands with *A. phagocytophilum* infection. While this enzyme is canonically known to modify and stabilize collagen [8], it has an uncharacterized function in tick saliva and tick salivary glands [133]. However, there is a high abundance of tick cement proteins with proline hydroxylation motifs [133], suggesting that *A*. *phagocytophilum* may be enriching for a subtype of cement proteins. Additionally, proyly hydroxylase-mediated upregulation of a putative ACE I in salivary glands may function to degrade bradykinin, which is involved in pain sensation and edema [20,134]. The overall effect of *A*. *phagocytophilum* manipulation of the tick sialome seems to enhance tick feeding success by increasing anticlotting activity as well as reducing pain sensation and edema. Additionally, *A*. *phagocytophilum* may selectively enhance a subpopulation of cement proteins with uncharacterized functional significance (Figure 2A).

*B*. *burgdorferi*, like *A*. *phagocytophilum*, increases the expression of several common anticlotting factors and cement proteins [136]. Anticlotting factors Salp11 and Metis-1 levels are increased during *Borrelia* infection compared to uninfected controls. Other proteins include a prolyl hydroxylase, also affected during *A*. *phagocytophilum* infection, suggesting that the exploitation of anticoagulants, and potentially collagen integrity (Figure 2B), benefits infection. Manipulation of the expression of salivary proteins appears to be a conserved mechanism used by *Borrelia* spp. bacteria. *Borrelia afzelii*, the main causative agent of Lyme disease in Europe, also alters the gene expression in the salivary glands of *I*. *ricinus* [137]. Using two different sequencing approaches to determine gene expression levels, Trentelman et al. [137] determined that *B*. *afzelii* had the biggest effect on salivary protein gene expression at 24 h of feeding, affecting 465 genes. These include genes encoding statins, immunity relate genes, signal transduction, ixodegrin family, Salp15 family, protein export, metalloproteases, lipocalins, and serine proteases. Lipocalins are a family of conserved proteins that bind to diverse targets, including histamine, Leukotriene B4 (LTB4), and others. *B*. *azfelii* infection increases the expression of a lipocalin, JAA67401, a putative serotonin binding protein [138]. Serotonin and the metabolites generated in the serotonin pathway are important in coordination between the neuroendocrine and immune systems. Serotonin, melatonin, and other neurometabolites are produced by immune cells and are substrates for enzymes that synthesize immunomodulators and anti-inflammatory molecules [139]. Whether other *Borrelia* species also manipulate these genes has not been tested. However, both *B*. *afzelii* and *B*. *burgdorferi* enhance the expression of Salp15 [123], indicating potentially conserved mechanisms to evade antibody recognition (Figure 2B). 

RNA-seq analysis of the salivary glands of TBEV-infected *I*. *ricinus* females attached for 1 or 3 h shows that TBEV infection increases the expression of anticlotting/immunomodulatory genes (lipocalins, metalloproteases, protease inhibitors, and lectins), and genes encoding cement proteins (four mucin genes and 11 genes for glycine rich proteins) [140]. In this study, the authors speculated that since most of the mucins and glycine rich proteins were upregulated within 1 hour of attachment, a potential shift in the composition of the cement cone that may occur compared to uninfected ticks. The importance of these proteins is highlighted by the protective effect that vaccination using a glycine rich protein, 64TRP, has against TBEV transmission to mice [141] (Figure 2C). 

Interestingly, *A*. *phagocytophilum*, *B*. *burgdorferi*, and TBEV, all manipulate gene expression in tick salivary glands, in particular those coding for anticlotting factors, immune inhibitors, and proteins involved in cement protein production. *A*. *phagocytophilum* and *B*. *burgdorferi* both upregulate levels of Salp11, Metis-1, and prolyl 4-hydroxylase. All three also increase the expression of genes encoding immunomodulatory proteins. While there are some data showing that cement protein expression changes during infection with all three pathogens, only the increase in glycine rich proteins in TBEV-infected salivary glands has some documented benefit for pathogen transmission. Together, these studies suggest that these pathogens may target similar effectors and pathways to enhance tick feeding success and vertebrate host immune evasion to facilitate their transmission.

## 5. Conclusions

To support the uptake of large volumes of blood over a long duration of time, ticks have evolved a large repertoire of salivary molecules to counteract host defense mechanisms. Tick saliva contains components that interfere with normal hemostasis and immunological mechanisms at the bite site. The immunomodulatory components in tick saliva create a privileged environment that can enhance the survival and transmission of tick-borne pathogens. In this context, it is not surprising that tick-borne pathogens direct changes in tick salivary glands and saliva to further their survival and transmission, by modifying gene expression and directly binding to salivary products. This review focused mainly on proteinaceous effectors. Nevertheless, tick-borne pathogens may enhance their transmission by exploiting other molecules. A recent study has shown that Powassan virus, a TBEV complex member, is able to alter the expression of miRNAs in the salivary glands of *I*. *scapularis* ticks [142], several of which represent novel miRNAs not previously reported, while others match previously identified sequences. These findings are corroborated by the in vitro transfection of monkey kidney (Vero) epithelial cells with inhibitors of some of these miRNAs before infection with the Powassan virus which resulted in either higher or lower viral loads. These results suggested that the secretion of these miRNAs in tick saliva may serve to limit infection at certain time points, while aiding at others. miRNAs have been detected in the saliva of other tick species [143,144]. Thus, other pathogens may also target miRNAs expression to facilitate their transmission. 

Another yet unexplored mechanism by which tick-borne pathogens may hijack the secretion of salivary effectors is the manipulation of extracellular vesicles. Extracellular vesicles are small lipid blebs that secrete for cell-to-cell communication. Extracellular vesicles have been detected in the saliva of *I*. *scapularis*, *Amblyomma maculatum*, and *H*. *longicornis* [145,146,147]. Vesicles from *H*. *longicornis* contain known protein effectors, like lipocalins, cement-like proteins, and serpins as well as novel miRNAs suspected to influence host immune responses [145,146]. Furthermore, in vitro experiments indicate that vesicles from *I*. *scapularis* and *A*. *maculatum* diminish chemokine and cytokine secretion and delay wound healing responses [147]. Several studies have shown their immunomodulating effects and influence on infection by several vector-borne pathogens [148], including the transmission of Langat virus (LGTV) from tick cells to mammalian cells [149]. Interestingly, LGTV was not only able to change the proteomic and genomic cargo of the vesicles by adding its own material, but also increased the number of extracellular vesicles secreted. It is possible that other tick-borne pathogens similarly influence the cargo and secretion of these vesicles to facilitate their transmission. 

A major goal of studying the tick-host-pathogen interface is the discovery of the genetic components and molecular pathways that contribute towards the transmission of tick-borne pathogens. While so far only a few factors and mechanisms have been identified, it is evident that tick-borne pathogens manipulate salivary gland components to enhance tick feeding success and their transmission to the host. Distinguishing and characterizing these immunomodulatory molecules could serve to identify potential targets for the development of future tick control measures and vaccine targets that could positively block tick-borne pathogen transmission. 

## Figures and Tables

**Figure 1 ijerph-18-01806-f001:**
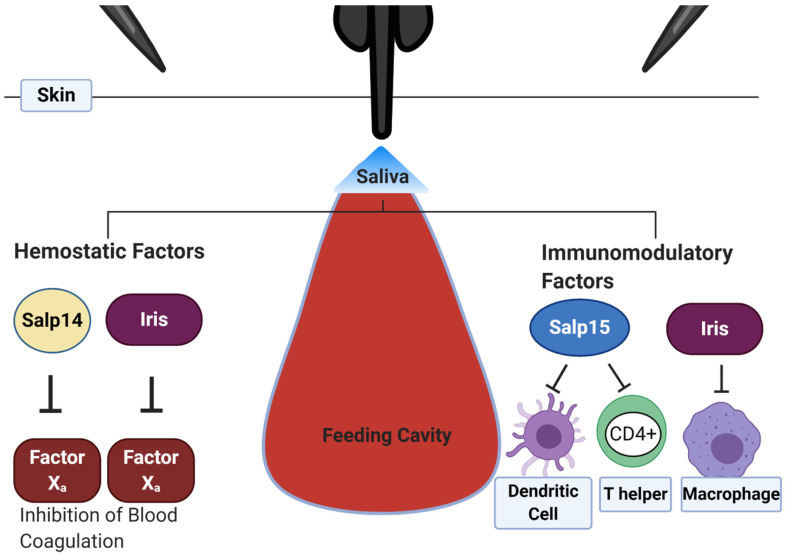
Secretion of anti-hemostatic (Salp14 and Iris) and immunomodulatory components (Salp15 and Iris) within tick saliva. Salp14 inhibits Factor Xa and the conversion of prothrombin to thrombin. Iris, an elastase inhibitor, hinders the intrinsic or contact-dependent coagulation pathway and platelet aggregation. The immunomodulatory protein, Salp15 prevents dendritic cell function by binding to the DC-SIGN (dendritic cell-specific intercellular adhesion molecule-3-grabbing non-integrin), a lectin receptor. This binding alters dendritic cell cytokine secretion. Additionally, Salp15 binds the CD4 glycoprotein on CD4+ T-helper cells, inhibiting the T-cell receptor signaling. Iris decreases the production and secretion of pro-inflammatory IL-6 and TNF-α by macrophages and affects T cell proliferation. Created with Biorender.com

**Figure 2 ijerph-18-01806-f002:**
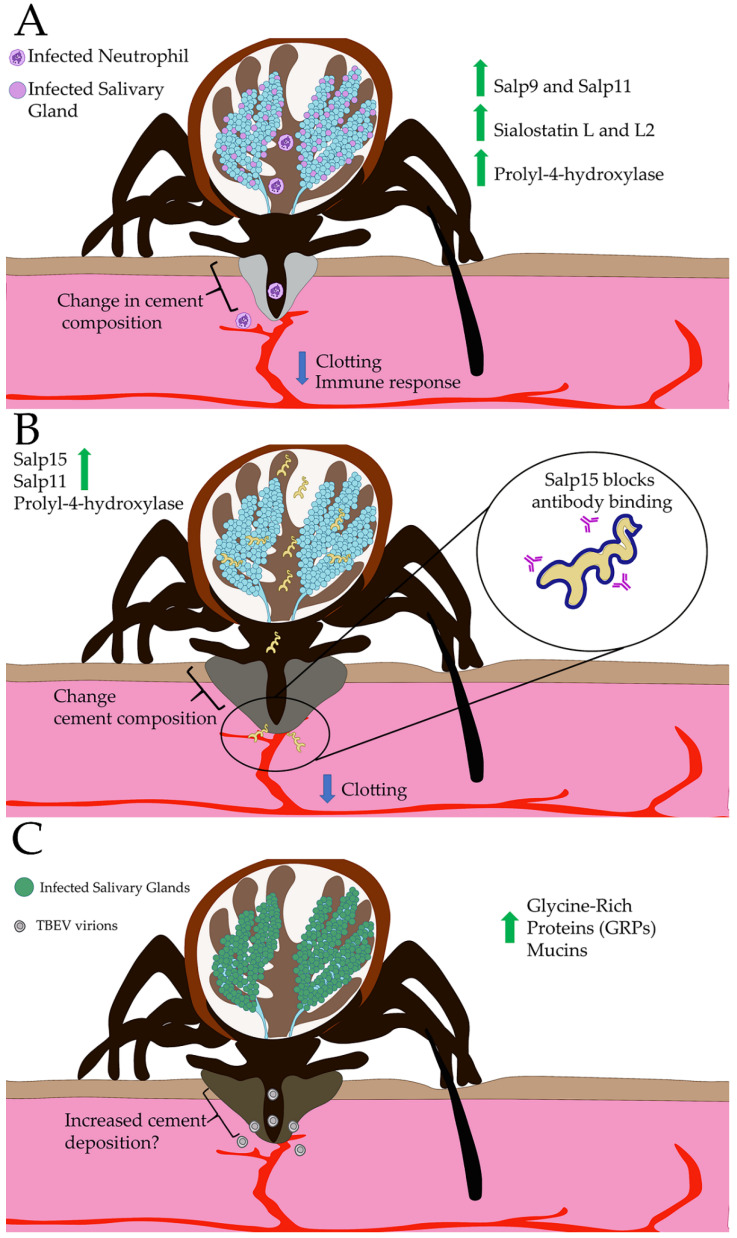
Tick-borne pathogens manipulate the expression of tick salivary effectors to enhance their transmission. (**A**) *A*. *phagocytophilum* infected neutrophils are taken up by the tick during its bloodmeal. Once in the tick, *A*. *phagocytophilum* infects salivary glands as early as 24 to 48 h. In the salivary glands, *A*. *phagocytophilum* leads to increased protein levels of the anticoagulants Salp9 and Salp11, the immune inhibitors Sialostatin L and L2, and a prolyl 4-hydroxylase. Sialostatin L and L2 are cystatins that bind and inhibit cathepsins L and S, leading to decreased and altered immune responses at the bite site. Prolyl 4-hydroxlases are enzymes that hydroxylate prolines and are necessary for collagen stability. Furthermore, the presence of proline hydroxylation motifs in several cement proteins suggests that *A*. *phagocytophilum* may influence the composition of the cement cone. (**B**) *Borrelia* pathogens selectively increases Salp15 expression in salivary glands. Salp15 directly binds OspC in the surface of *Borrelia burgdorferi*. The Salp15 coating blocks OspC specific antibodies from binding the bacteria, inhibiting both direct antibody mediated killing and activation of complement pathways. Salp11 and prolyl 4-hydroxylase proteins levels are increased during *Borrelia* infection, delaying clot formation and potentially affecting collagen stability. Similarly to *A*. *phagocytophilum,* the effect of *B*. *burgdorferi* infection on prolyl 4-hydroxylase expression may indicate that bacterial infection leads to modifications in the composition of the cement cone. (**C**) TBEV infection alters the expression of genes encoding several cement proteins, mucins, and glycine-rich proteins. The increased expression of glycine-rich proteins may represent an overall increase in tick cement proteins or a compositional shift of cement proteins being deposited to form the cement cone. This compositional change may be necessary for TBEV transmission as vaccination with a glycine rich protein resulting in protection against virus infection.

**Table 1 ijerph-18-01806-t001:** Characterized anti-hemostatic components secreted in tick saliva *.

Component	Function/Activity	Target Effector/Protein	Pathogen SAT	Tick Species	References
Apyrase	Inhibitor of platelet aggregation	Adenosine triphospahte (ATP), Adenosine diphosphate (ADP)		*Ixodes scapularis*, *Ornithodoros savignyi*	[73,75]
Tick histamine release factor (tHRF) †	Binding to basophils, stimulates Histamine release, vasodilation	-	*B. burgdorferi*	*Dermacentor andersoni*, *I. scapularis*	[49,69]
Metalloproteases	Wound healing/angiogenesis	Fibrin		*Ixodes ricinus*	[56]
Haemangin	Wound healing/angiogenesis	Trypsin, chymotrypsin, plasmin		*Haemaphysalis longicornis*	[62]
HLTnl	Wound healing/angiogenesis	Competitive inhibitor of Vascular endothelial growth factor (VEGF) for VEGF receptor		*H. longicornis*	[60]
PGE2 †	Wound healing/angiogenesis	PGE_2_ receptor, cyclic Adenosine monophosphate (AMP)-proteins kinase A (cAMP-PKA)		*D. variabilis*	[72]
TAP	Blood coagulation	Factor Xa		*Ornithodoros moubata*	[79]
Moubatin	Platelet activation and vasoconstriction inhibitor	Collagen-stimulated aggregation activator		*O. moubata*	[68,78]
Disaggregin	Platelet aggregation	Platelet fibrinogen receptor		*O. moubata*	[63]
Enolase	Blood coagulation	Fibrin, plasminogen receptor		*O. moubata*	[58]
Savignygrin	Platelet Aggregation	Thrombin		*Ornithodoros savignyi*	[67]
Longicornin	Platelet aggregation	Collagen		*H. longicornis*	[54]
Ornithodorin	Blood coagulation	Thrombin		*O. moubata*	[76]
Salp14	Blood coagulation	Factor Xa		*I. scapularis*	[47]
Variabilin	Platelet aggregation	Glycoprotein IIb-IIIa		*D. variabilis*	[77]
Serpin19	Blood coagulation	Factor Xa, factor XIa, trypsin, plasmin		*Amblyomma americanum*	[64]
RmS-15	Blood coagulation	Thrombin		*Rhipicephalus (Boophilus) microplus*	[80]
Longistatin	Blood coagulation	Fibrin		*H. longicornis*	[50]
IxscS-1E1	Blood coagulation	Thrombin, trypsin		*I. scapularis*	[61]
IRS-2 †	Blood coagulation	Inhibits Cathepsin G and chymase proteases		*I. ricinus*	[55]
Ir-CP1	Blood coagulation	Inhibitor of contact system proteins		*I. ricinus*	[57]
Variegin	Blood coagulation	Direct competitive inhibitor of Thrombin		*Amblyomma variegatum*	[65,66]
Amblyomin-X	Blood coagulation	Noncompetitive inhibitor of coagulation factor, Factor Xa		*A. variegatum*	[51,52]
Ixolaris	Blood coagulation	Inhibitor of contact system proteins, binds to Factor Xa		*I. scapularis*	[53,59,70]
Iris †	Blood coagulation	Thrombin, Factor Xa, tissue plasminogen activation inhibitor		*I. ricinus*	[48]
Savignin	Blood coagulation	Thrombin		*O. savignyi*	[71]
TSGP3	Platelet aggreagation and vasoconstriction inhibitor	Inhibition Collagen-platelet binding and interaction with thromboxane A2		*O. savignyi*	[68]
TIX-5	Blood coagulation	Factor Xa, factor V		*I. scapularis*	[74]

†: Denotes as having immunomodulatory function as well. * adapted and updated from [3,19,25,81].

**Table 2 ijerph-18-01806-t002:** Described immunomodulatory components in tick saliva *.

Component	Function/Activity	Target Effector/Protein	Cells Affected	Pathogen	Tick Species	References
† Iris	Th 2 response modulation	Reduction in TNFα, INFγ, IL-8, IL-6, and IL-1β expression	Macrophages, T-lymphocyte		*I. ricinus*	[97]
Salp15	IL-2 inhibitor, T-cell proliferation (*I. scapularis*), IL-10 secretion inhibitor (*I. ricinus*)	OspC	T cells and peripheral blood mononuclear cells (PBMCs)	*B. burgdorferi*	*I. scapularis*, *I. ricinus*	[86,91,94,99,100]
IL-2 Binding Protein	T cell proliferation	IL-2	T cells and PBMCs		*I. scapularis*	[16]
IR-LBP	Neutrophil migration	Leukotriene B4	Neutrophils		*I. ricinus*	[86]
Irac I & II	Complement inhibitor	C3 convertase			*I. ricinus*	[88]
Isac	Complement inhibitor	C3 convertase			*I. scapularis*	[104]
Salp16 Iper1	Neutrophil migration and reactive oxygen species (ROS) inhibitor		Neutrophils	*A. phagocytophilum*	*I. persuculatus*	[21]
Salp16 Iper2	Neutrophil migration and ROS inhibitor		Neutrophils	*A. phagocytophilum*	*I. persculutaus*	[21]
† IRS-2	Immune inhibitor	IL-6, IL-9, and IL-17 secretion STAT-3 phosphorylation	Dendritic cells, Neutrophils, and Th17 cells	*B. burgdorferi*	*I. ricinus*	[23]
Sialostatin L and L2	Immune inhibitor	Cathepsin L and S inhibitor, inflammasome formation		*B. burgderfori*, *L2-TBEV* and *A. phagocytophilum*	*I. scapularis*	[38,41,42]
Japanin	Modulates DC maturation	Inhibits IL-1β, IL-6, IL-12, IFN-γ, and TNFα secretion, CD86 and CD83 expression. Enhances IL-10 secretion and CD274 expression	Dendritic cells		*R. appendiculatus*	[24]
IrSPI	T cell proliferation	CXCL10, CCL7, CCL4, CCL5, Eotaxin, IFN-γ, IL-1β, IL-18, IL-13, IL-6, TNFα, IL-9, and Granulocyte macrophage-colony stimulating factor (GM-CSF) inhibition and IL-2	CD4^++^ T cells		*I. ricinus*	[87]
† PGE_2_	Immune inhibitor	Inhibition of IL-12, TNFα, and CD40 and upregulation of IL-10 (*I. scapularis*), increased macrophage PGE_2_, CCL5, TNFα, and sTNFRI secretion (*D. variabilis*), and TNFα inhibition (*A. sculptum*)	Bovine mononuclear cells, DCs, and macrophages	*Rickettsia rickettsii*	*R. (Boophilus) microplus*, *I. scapularis*, *D. variabilis*, and *Amblyomma sculptum*	[72,90,95,101]
Macrophage Migration Inhibitory Factor (MIF) homolog	Macrophage migration		Macrophages		*A. americanum*	[22]
BIP (B-cell inhibitory protein)	B cell proliferation		B cells	*B. burgdorferi*	*I. ricinus*	[91]
B-cell inhibitory factor (BIF)	B cell proliferation		B cells		*Hyalomma asiaticum*	[105]
Amregulin	Immune inhibitor and antioxidant	TNFα, IFN-γ, IL-1, IL-8, and Nitric Monoxide (NMO) inhibitor			*A. variegatum*	[102]
tHRF	Histamine release	Histamine release stimulation	Basophils	*B. burgdorferi*		[49]
TSGP2/3	Neutrophil migration and complement inhibition	Leukotriene B4 and C5 binding			*O. savignyi*	[68]
Salp20	Complement inhibition	C3 convertase (from properdin displacement)		*B. burgdorferi*	*I. scapularis*	[93,96,104]
Iristatin	T cell proliferation and immune inhibition	IFN-γ, IL-2, IL-4, IL-6, and IL-9 secretion, CD4^+^ T cell proliferation, neutrophil migration, and nitric oxide production	Neutrophils, macrophages, T and mast cells		*I. ricinus*	[96]
DsCystatin	Immunomodulation	Cathepsin L and B inhibitor, TNFα, IL-6, IL1β, and IFNγ inhibition, and promotes TRAF6 degradation	Macrophages	*B. burgdorferi*	*Dermacenter silvarum*	[101]

†: Denotes as having anti-hemostatic function as well. * adapted and updated from [3,19,25,81].

## Data Availability

No new data were created or analyzed in this study. Data sharing is not applicable to this article.

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
