# Peer review of "Changing the Recipe: Pathogen Directed Changes in Tick Saliva Components"

_ijerph, 2021, doi:10.3390/ijerph18041806_

Round 1

Reviewer 1 Report

This manuscript is a review of literature on the issue of tick-borne pathogens' modification of the human immune system to enhance pathogen success.  In light of recent emergence of tick-borne diseases, it is timely.   It is also well-written and interesting.  My comments are relatively minor and should be accommodated easily by the authors.

In line 216:   the authors mention that some animals may have developed immunity to tick feeding.  It would be useful to know what animals have developed such immunity.   If it has not been well demonstrated, that should be mentioned in the text.

Line 264:  remove "m" from "influences"

Line 277 wording "comprises of...."

Line 298 "proyly"  check spelling 

line 302:  remove "and"

line 319:  "Thus, inhibiting direct antibody..."    This is not a sentence.   Connect it to the previous sentence.

line 392:  Change "my" to "may".

I think this is an important review and should be accepted for publication with minor revisions.

Author Response

Please find the revision of our manuscript entitled "Changing the Recipe: Pathogen directed changes in Tick Saliva Components" by Michael Pham, Jacob Underwood, and Adela Oliva Chavez attached. We are grateful to the reviewers for their kind and thoughtful review. We have address them completely, with the exception of some comments which may affect the formatting requirements by the journal. In those cases, we have replied point by point with the reasons why we did not perform the changes suggested.

All the changes have been highlighted in yellow or with track changes.

Please find our response to each reviewer changes below:

Reviewer #1

Comment 1:In line 216:   the authors mention that some animals may have developed immunity to tick feeding.  It would be useful to know what animals have developed such immunity.   If it has not been well demonstrated, that should be mentioned in the text.”

We have added the specific animal models which have shown the development of immunity against salivary secretions. These species include rabbits, guinea pigs, and cattle. This description can be found in lanes 223-225.

Comment 2: “Line 264:  remove "m" from "influences".”

We have removed the m.

Comment 3:Line 277 wording "comprises of....".”

We have changed the working to “consists of”.

Comment 4: “Line 298 "proyly" check spelling”.

We have modified the spelling to “proline hydroxylation motifs”.

Comment 5: “line 302:  remove "and"”.

We have removed “and” as requested. It now reads: “as well as reducing”.

Comment 6: “line 319:  "Thus, inhibiting direct antibody..."    This is not a sentence.   Connect it to the previous sentence.”

We have connected both sentences. It now reads: “The Salp15 coating blocks OspC specific antibodies from binding the bacteria, inhibiting both direct antibody mediated killing and activation of complement pathways.”.

Comment 7: “line 392:  Change "my" to "may".”

The word has been changed as requested.

Reviewer #2

Comment 1: “2.1. Saliva Assisted Transmission (SAT) Is SAT observed only during vertebrate infection by tick-borne pathogens? Is the pathogen transmission from the infected vertebrate host to feeding tick facilitated? Please add some information”.

In fact, SAT has been reported in other systems. We have added additional information in lanes 78-82. Some studies do, in fact, indicate that tick saliva may facilitate the chemoattraction of Anaplasma phagocytophilum infected neutrophils and the extracellular bacterium Borrelia burgdorferi. The exact mechanism that induce this chemoattraction are still unknown. We describe these studies in lanes 83 – 93.

Comment 2: Lines 105-112 Reference [35] is added twice. It should be removed to the end of the paragraph.”

We have removed the extra reference and it is only included at the end of the paragraph as requested by the reviewer.

Comment 3: The figures contained in the manuscript are very aesthetic and useful, however, the quality of the Figure 1 should be improved (descriptions should be larger, clearer). Secondly, for better explenation, you can mark in the picture which part of the figure is the saliva, tissue etc.”

We thank the reviewer for this comment. We have modified the figure and we consider that it is improved.

Comment 4: “Lines 144-147, 273-275 The same mistake as Lines 105-112 - Reference is added twice. It should be removed to the end of the paragraph.”

We have removed the double citation as requested.

Comment 5: Lines 195, 215, 239, 261, 286, 304, 371 - References should be moved or added at the end of the paragraph.

There are no references associated with those statements as they represent either the speculations or hypothesis of the authors. By adding references, we would be misleading the readers.

Comment 6: All References included in Tables should be included in the text first (preferably precede the table).”

The references for table 1 have been added in line 145 and in line 177 for table 2.

Comment 7: “The reference list should be reviewed for errors and journal requirements (uppercase, lowercase etc.)”

We have revised every reference and have made the necessary changes to meet the journal requirements.

Again, we really appreciate the comments from the reviewers. The quality of the manuscript has improved.

Sincerely,

Adela Oliva Chavez

Assistant Professor

Department of Entomology

Reviewer 2 Report

The review in the file below

Author Response

(The authors gave the same response as above.)
